# Vitellogenesis and Embryogenesis in Spiders: A Biochemical Perspective

**DOI:** 10.3390/insects16040398

**Published:** 2025-04-10

**Authors:** Carlos Fernando Garcia, Aldana Laino, Mónica Cunningham

**Affiliations:** Instituto de Investigac iones Bioquímicas de La Plata “Prof. Dr. Rodolfo R. Brenner” (CONICET-UNLP), La Plata, Buenos Aires 1900, Argentina; aldana_laino@hotmail.com (A.L.); cunninghammoni@gmail.com (M.C.)

**Keywords:** spider, vitellogenesis, embryogenesis

## Abstract

This work reviews all currently available information on two fundamental processes in spider reproduction: vitellogenesis and embryonic development. In oviparous animals, resource accumulation in eggs occurs through vitellogenesis, and embryos consume these energy reserves during embryonic development. In spiders, knowledge of these processes is limited and has never been comprehensively reviewed. Recent studies have identified various lipoprotein particles in the ovaries, hemolymph, and yolk. The structure of lipovitellins and vitellogenins (lipoproteins involved in reproduction) has been compared across different organs and spider species. Structural lipids, mainly phosphatidylcholine and phosphatidylethanolamine, were the predominant components of eggs, followed by triacylglycerols. Additionally, for the first time, reproductive indices (gonadosomatic and hepatosomatic) are described, providing a new tool for studying vitellogenesis. Hemocyanin (a hemolymphatic protein) was detected in eggs during early stages, suggesting a role in organ formation. Furthermore, continuous lipovitellin consumption was observed throughout embryonic development until juvenile emergence.

## 1. Introduction

Spiders (Order Araneae) constitute the most diverse and species-rich group of the Class Arachnida. To date, more than 52,000 species have been recorded worldwide [1], inhabiting all accessible environments except for the open sea and the air. Most spiders live far from human settlements, though some species are frequently encountered in domestic environments.

Over time, numerous studies have investigated spider reproduction. The vast majority have focused on reproductive biology, analyzing aspects such as courtship behavior, sexual selection, and copulation [2,3,4,5,6]. However, the study of the biochemistry of vitellogenesis and embryogenesis has long been overlooked, despite their crucial role in the reproduction of the Order Araneae.

Oviparous animals, including spiders, store in the yolk of their eggs all the resources necessary for embryo development and the survival of larval stages until the offspring can feed on their own [7,8]. This yolk consists of various biomolecules that provide calories and energy resources, such as lipids, proteins, and carbohydrates. Additionally, it contains biomolecules with different functions, including pigments, defense proteins, and respiratory proteins such as hemocyanin (Hc).

The process of yolk formation, known as vitellogenesis, is the central event in egg formation for all oviparous animals. It is a seasonal or cyclic phenomenon during which the components of the yolk are stored in an organized manner within the oocyte [9,10,11], which increases in size due to the accumulation of lipid droplets, proteins, and carbohydrates [9,10,12,13,14,15].

Although some key compounds involved in vitellogenesis have been identified [16,17,18], knowledge of this process in spiders remains limited. To date, only ecdysteroids and juvenile hormones have been recognized as regulators of vitellogenesis. Ecdysteroids appear to play a predominant role in midgut gland vitellogenesis, while juvenile hormone seems to be more relevant in the ovaries [16]. However, further studies are needed to fully understand their specific roles and interactions. Most available information still comes from structural studies of eggs and ovaries [15,19,20,21,22,23].

## 2. Vitellogenesis in Spiders

### 2.1. Vitellogenins and Lipovitellins: Protein Composition

During vitellogenesis, yolk precursor proteins (YPPs), which have been of interest in studies since the 1980s, are produced in large quantities by females during reproductive stages and eventually become part of the eggs. Unlike the other two major macronutrients provided by females to the eggs (lipids and carbohydrates), proteins are not stored in the organisms, so they must be expressed when needed, using precursors (amino acids) primarily acquired from the diet. Among these proteins are vitellogenins (Vg), which are the most physiologically relevant lipoproteins for oviparous reproduction.

Vg is a hemolymphatic lipoprotein exclusive to females in the vitellogenic state. It was originally described as a sex-linked lipoprotein to differentiate it from the hemolymphatic lipoproteins of males and non-vitellogenic females. Vgs transport lipids and proteins exclusively to the ovaries, while the lipoproteins common to both sexes transport lipids from sites of absorption, storage, and synthesis to the various target organs where they will be utilized [24,25,26,27]. The synthesis and accumulation of Vg during the vitellogenic process in arthropods are well described in insects, crustaceans, and ticks. Generally, this process occurs in three stages: (1) synthesis of Vg (with post-transcriptional and post-translational modification) and release into the hemolymph (HL), (2) transport of Vg to the ovaries, and (3) uptake of Vg mediated by the ovarian receptor [8,28,29,30,31,32].

The synthesis of Vg is complex and varies among different animal groups. Generally, exogenous (heterosynthesis or extraovarian), endogenous (autosynthesis or ovarian), and, in some arthropod groups, a combined synthesis of both are distinguished [24,33,34]. Regarding extraovarian synthesis, different synthesis organs have been reported in arthropods: in crustaceans, it mainly occurs in the hepatopancreas (although it has also been described in various organs), in insects in the fat body, in ticks in the midgut and fat body, and in spiders in the midgut diverticula (MD) [16,18,24,35,36,37,38,39,40,41].

The term lipovitellin (LV) often refers to the Vg endocytosed by the ovary; therefore, it is common for them to share apolipoproteins, immunological identity, and amino acid sequences [42,43,44]. In some arthropods, it has been determined that Vg undergoes modifications in the ovary to form LV, which is later found in eggs, while in others, this modification continues until the formation of the so-called vitellins (Vn) [45,46,47].

In the vitellogenic process, it is common for the apolipoproteins of Vg to be restructured after proteolytic cleavage into LVs [48]. In the insect *Locusta migratoria*, where sequence homology between Vg and LV was observed, there was a variation in lipid composition, which is likely due to the contribution of ovarian lipids [49,50,51,52].

The information and nomenclature used for arthropods by different researchers is often confusing, as LV is frequently defined as Vn. Additionally, the various purification methodologies complicate generalizations regarding lipoprotein particles or their apolipoproteins [53]. Most studies on vitellogenesis in spiders, as in other arthropods, have focused solely on the protein component, either through amino acid sequence alignment, immunological identity, or comparison of molecular weights of proteins.

Table 1 lists the molecular weights of Vg or Vn proteins from spiders, along with some examples from mites. It shows that *Tegenaria atrica* has a 47 kDa protein in hemolymph and ovary with Vg immunological identity, matching a protein of the same molecular weight in *Parasteatoda tepidariorum*, detected in hemolymph, ovaries, and midgut glands [16]. Additionally, in *P. tepidariorum*, proteins of 250 and 30 kDa have been described. Using anti-LV polyclonal antibodies, four proteins (116, 87, 70, and 42 kDa) were identified in *Pardosa saltans* [54].

Although there are many spider species, to date, lipoprotein particles (LVs) related to reproduction have been purified and characterized in only two species, *Schizocosa malitiosa* and *Polybetes pythagoricus*. These LVs were isolated from eggs using density gradient ultracentrifugation, as, like all lipoproteins, they have both a protein component and a lipid component, which determines their hydration density (see below). In *S. malitiosa*, SmLV1 contains three distinct proteins, while SmLV2 contains four. *P. pythagoricus* has LV1 with four proteins and LV2 with six.

In the case of mites, the number of different proteins is higher than in spiders, possibly due to a greater number of cleavage sites or a higher contribution of proteins from different origins. Reported proteins range from 6 to 13, with molecular weights between 3.6 and 290 kDa.

Table 2 presents the N-terminal amino acid sequences of the proteins corresponding to the 75, 67, 46, and 30 kDa bands obtained from electrophoresis of a *P. pythagoricus* egg homogenate in a 4–23% polyacrylamide gradient, as well as the 47 kDa band from *T. atrica*. Additionally, the table includes references to the Vg genes of *Pardosa pseudognnulata* and *P. tepidariorum*.

To date, two proteins associated with Vg have been identified in different spider species: a 30 kDa proteins, present in *P. tepidariorum* (described as Vg) and in LV1 and LV2 of *P. pythagoricus*, and a 47 kDa protein, found in *P. tepidariorum*, *T. atrica*, and LV1 of *P. pythagoricus* (as 46 kDa).

Notably, the sequence of the 67 kDa protein showed 77% similarity to subunit 6 of the Hc of the spider *Cupiennius salei* (accession No. CAC44757.1), while the 47 kDa protein from *T. atrica* revealed 66% sequence similarity to the Vg I precursor of the fish *Fundulus heteroclitus* (accession No. Q90508) [18,56].

Since females synthesize, transport, and accumulate nutrients in the eggs during vitellogenesis, one way to study this process is by analyzing changes in the ovaries, hemolymph, and certain extraovarian Vg synthesis organs.

The gonadosomatic index (GSI) is the most widely used mathematical expression to describe gonadal development in various arthropods. It is calculated as gonad mass (g) × 100/body mass (g). Initially applied to fish [71,72,73], it was later adapted for mollusks [74,75,76], insects [77,78], crustaceans [79,80,81], and for the first time in spiders in 2018 [10].

On the other hand, the hepatosomatic index (HSI) has been used in various invertebrates possessing a hepatopancreas, an organ comparable to the MD of spiders. Since there is no equivalent term to HSI for spiders, the same nomenclature has been adapted. Thus, HSI is calculated as MD mass (g) × 100/body mass (g) [10].

In *P. pythagoricus*, variations in these indices have been reported across different female reproductive stages (previtellogenic, early vitellogenic, vitellogenic, and postvitellogenic) (Figure 1). No significant changes were observed in HSI across the four stages, while GSI increased 2.9-fold during the early vitellogenic stage and a further 1.4-fold during the vitellogenic stage [10].

Although it is now accepted that Vg synthesis in spiders occurs in the MD [16] (see below), it is reasonable that no substantial accumulation of Vg is observed in this organ, leading to significant variations in MD mass beyond the general increase in body mass. This is because Vg is released into the hemolymph and subsequently endocytosed by the ovary, a conclusion supported by the commonly observed increase in total protein content in hemolymph. A similar pattern has been reported in crustaceans, such as *Penaeus schmitti* [82] and *Litopenaeus merguiensis* [83].

Conversely, in *P. tepidariorum*, a strong correlation was found between Vg concentration in the MD and reproductive status [16]. Additionally, Romero et al. [56] reported an increase in total protein content in MD of vitellogenic *P. pythagoricus* females.

### 2.2. Lipid Composition and Yolk

As previously mentioned, during vitellogenesis, the macronutrients (proteins, carbohydrates and lipids) required for embryonic development are stored within the oocytes [9,11]. In oviparous animals, these lipids are components of membrane structures and lipid droplets, as well as of the characteristic yolk lipoproteins mentioned earlier [84].

During the reproductive state, females exhibit high lipid concentrations in their synthesis or storage organs, hemolymph and ovary. In crustaceans and scorpions, lipids are stored in the hepatopancreas, whereas in insects, they accumulate in the fat body [36,85,86,87,88].

In spiders, in vivo and in vitro assays using radiolabeled lipids have determined that the MD serves as the primary organ for lipid metabolism and storage [26,27].

The variation in lipid dynamics in the MD and ovary of spiders has been studied during the early stages of vitellogenesis, specifically in the previtellogenic stage [10] (Figure 1). The increase in lipids within the MD is associated with their subsequent accumulation in the ovary, as described for other arthropods [89,90].

Additionally, part of this lipid increase in the MD may serve to support the female’s metabolism during oviposition and the subsequent maintenance of the egg sac, both of which are highly energy-demanding processes [91]. In *P. pythagoricus*, as in other spiders, females feed very little after oviposition, devoting themselves entirely to egg sac care and fiercely defending it [92]. Similarly, in the shrimp *Penaeus monodon*, it has been suggested that lipid reserves accumulated in the hepatopancreas are essential to meeting energy demands during and after oviposition [93].

The lipids accumulated in the ovaries of vitellogenic females [94,95] contribute to the increase in oocyte size [96,97]. In the spider *P. pythagoricus*, ovaries in the vitellogenic stage were found to contain a high proportion of energetic lipids (23% triacylglycerols (TAGs)) and a substantial amount of structural lipids (65% phosphatidylethanolamine (PE) + phosphatidylcholine (PC) + sphingomyelin (SM)) [10]. A similar lipid profile was observed in the ovarian lipids of the arthropod *P. monodon* [93].

It is important to highlight that while PC is the predominant structural lipid in the hemolymph lipoproteins of several non-reproductive spiders [98,99,100], during vitellogenesis in *P. pythagoricus*, PE showed a significant increase in both the hemolymph and vitellogenic ovaries [10]. This makes PE the main structural lipid associated with spider reproduction, which is also consistent with its presence in LV [57].

The literature describes how different diets influence the fatty acid (FA) composition in arthropods [101,102,103]. In *P. pythagoricus* females, as well as in scorpions, tarantulas, and other spiders in a non-reproductive state, no major changes in FA composition have been observed despite dietary differences. The predominant FAs are 18:1 and 18:2, with lower proportions of 18:0 and 16:0, which follows the same pattern observed in whole-body samples and MD of other three species of labidognath spiders [26,104] and in the hepatopancreas of scorpions [104,105].

In the MD of *P. pythagoricus*, an enrichment of 18:2 and a depletion of 16:0 were observed compared to their respective levels in the hemolymph during previtellogenesis. This pattern matches the one described in scorpions by [105], where the authors suggested that the hepatopancreas contained more than one FA group with different levels of exchange with the hemolimph. A similar situation was later observed in other scorpions [106].

During vitellogenic development in spiders, hemolymph proteins maintain a constant concentration during both the previtellogenic and early vitellogenic stages but show a significant increase (45%) during the vitellogenic stage. After oviposition, hemolymph protein content decreases, reaching levels similar to those found in postvitellogenic, non-vitellogenic females and males.

When the Hc content in the ovaries was analyzed throughout vitellogenesis, its accumulation in vitellogenic ovaries was observed. In this stage, Hc levels increased 80-fold compared to postvitellogenic or previtellogenic ovaries. Moreover, the high Hc content in vitellogenic ovaries was later found to be incorporated into the egg [56]. Although the origin of Hc in spider eggs has not been determined, it is possible that the ovary sequesters this protein through mechanisms similar to those reported in other arthropods, such as pinocytosis [107] or endocytosis [108].

In spiders, Vn accumulation occurs in two stages: the first in the young oocyte and the second after fertilization, provided that sufficient food is available [109].

Regarding the composition of this Vn, the presence of two LV, LV1 and LV2, has been described in *P. pythagoricus*, with densities of 1.16 and 1.23 g/mL, respectively. These LVs contribute 24.3 µg of protein per mg of egg, representing 27.8% of the total proteins [57,110]. Similarly, in *S. malitiosa*, two LVs, SmLV1 and SmLV2, have been identified with densities of 1.13 and 1.24 g/mL, providing 7.2 µg of protein per mg of egg, which accounts for 57.1% of the total proteins [55]. This pattern is comparable to that observed in *P. saltans*, where LV contributes 7.7 µg of protein per mg of egg, representing 35% of the total proteins.

In some spider families, eggs exhibit a typical structure known as the vitellin body or Balbiani body [111]. Initially, it was described as a yolk-organizing center [20], although its function remained unknown until structural and histochemical analyses were conducted on the oocytes of *Clubiona* sp. [21]. These analyses revealed that, during the early stages of oogenesis, the Balbiani body consists of two regions: a central core and a cortex. The central region is composed of filaments, mitochondria, and annulate lamellae, while the cortical zone primarily contains mitochondria. Additionally, the Balbiani body is consistently associated with elements of the rough endoplasmic reticulum and Golgi complexes. Histochemical studies showed that, during vitellogenesis, numerous lipid droplets form within the Balbiani body cortex [21]. Based on these findings, the authors proposed that one of the functions of the Balbiani body is the formation and accumulation of lipids used during embryonic development [112].

Since lipids represent one of the main energy sources, the lipid content of both the total yolk and yolk-associated lipoproteins (LVs) has been described in the spider species *S. malitiosa*, *P. pythagoricus*, and *P. saltans*. In the lycosid *S. malitiosa*, it was reported that of the total lipid content in eggs (8 mg/g of egg), 24.3% is contributed by LVs, with 19.8% corresponding to SmLV1 (1.6 mg/g of egg) and 4.5% to SmLV2 (0.37 mg/g of egg) [55]. Similarly, in *P. pythagoricus*, the total lipid content in eggs was 50 mg/g wet weight, of which LVs accounted for 28.9%, with 26.4% contributed by LV1 (13.24 mg/g wet weight) and 2.42% by LV2 (1.21 mg/g wet weight).

Table 3 compiles the major lipids found in LVs and spider eggs, as well as those present in other arthropods. In *S. malitiosa*, the predominant lipids in the egg cytosol are TAG and the phospholipids PC and PE, while in lipoproteins, the majority of lipids correspond to SM and lysophosphatidylcholine (LPC) [55]. In *P. pythagoricus*, the main cytosolic lipids are also TAG, PC, and PE; however, unlike *S. malitiosa*, its LVs contain a high proportion of esterified sterols (ESs), with 16.6% in LV1 and 24.2% in LV2. In *P. saltans*, the major lipids in egg extracts are TAG and phospholipids, primarily PC. The lipid extracts from eggs of the three spider species analyzed exhibit a similar composition, dominated by TAG and phospholipids, consistent with findings in other arthropods [25,113,114,115,116]. These two lipid classes are essential for organogenesis, as they contribute to membrane formation and serve as an energy source [117].

**Table 3 insects-16-00398-t003:** Main percentages (%) of the different lipids present in the LV and eggs of spiders, crustaceans and insects.

	LV1	LV2	Egg	References
*P. pythagoricus* (spider)	TAG: 8 CHOL: 8 ES: 16.6 PE: 25.4 PC: 23.8	TAG: 9.5 CHOL: 3.3 ES: 24.2 PE: 20.1 PC: 17.5	TAG: 22.9 PE: 48 PC: 22	[57,110]
*S. malitiosa* (spider)	TAG: 3.1 PE: 16.9 PC: 5.4 SM + LPC: 72.6	TAG: 0.4 PE: 3.6 PC: 0.8 SM + LPC: 98.1	TAG: 25.7 PE: 15.0 PC: 32.6 SM + LPC: 18.9	[55]
*P. saltans* (spider)			TAG: 45.9 PE: 3 PC: 28.4 LPC:5.5	[54]
*Alpheus saxidomus* (crustacean)			TAG: 51.1 PL: 48.9	[118]
*Palaemonetes schmitti* (crustacean)			TAG: 36.1 PL: 63.6	[118]
*Macrobrachium borellii (crustacean)*	TAG: 20.5 PC: 41.9 PE: 15.8		TAG: 55.4 PE: 13.2 PC: 14.7	[117,119]
*Locusta migratoria* (insect)	TAG: 0.4 PC: 62.1 PE: 21.8			[120]

TAGs are typically found in the yolk in the form of lipid droplets [121,122] and are the most important energy molecules due to their high caloric capacity and storage efficiency [21,55,57].

Among the polar lipids present in egg extracts are PC, LPC, PE, and SM. PC and SM are essential components of biological membranes and lipoproteins [123,124]. In *P. pythagoricus*, a notably high percentage of PE has been reported, similar to that found in the lipoproteins of the same species (see below) and in crustaceans [125]. PE has been proposed as a unique structural component necessary for the formation of inner mitochondrial membranes [10,126,127,128]. However, it may also have an energy-related role, as its consumption during embryonic development has been suggested by some authors for structural lipids [129,130] and described in certain crustaceans [131]. Additionally, we cannot rule out a regulatory function for PE in vitellogenic ovaries, as recently proposed for alkenyl-PE in the crustacean *Scylla paramamosain* [132].

On the other hand, in *S. malitiosa*, SM + LPC accounts for 18.9% of the total egg lipids. SM has been associated with several embryonic development functions, such as cell cycle arrest [133], stimulation of inositol phosphate production [134], cell proliferation and differentiation, and cellular membrane trafficking, among others [135,136].

Hcs are primarily known for their role as respiratory pigments in arthropods and mollusks. However, these molecules perform a wide range of additional functions, including contributions to homeostasis, immunity (through phenoloxidase activity and antimicrobial peptide formation), hormone transport, osmoregulation, and lipid transport [26,98,137,138,139,140,141,142,143,144,145].

The presence of this protein in arthropod eggs has been reported in insects [146,147], myriapods [148], and crustaceans [149]. However, available information on its presence in egg of spiders and chelicerates in general remains very limited. The first and only report identifying an Hc monomer (67 kDa) in spider LV was published in 2019. Its identity was confirmed through antibody detection and N-terminal sequence analysis (Table 1 and Table 2) [56]. This protein is found at a concentration of 10 mg per ootheca, representing approximately 15% of the total yolk proteins in newly laid eggs [56].

In spiders, it has been reported that cyanocytes and other types of hemocytes are responsible for the production and storage of Hc [109,150,151,152,153]. This synthesis is highly active, as Hc is the predominant circulating protein in the HL of spiders [154].

In *P. pythagoricus* [98], as well as in *Latrodectus mirabilis* [99] and *Grammostola rosea* [141], Hc is associated with different lipoprotein particles, including HDL and VHDL. In *P. pythagoricus*, Hc-containing lipoproteins account for 99% of the circulating lipoproteins, playing a major role in the lipid transport system [98,99].

## 3. Embryonic, Post-Embryonic Development, and Yolk Consumption

In oviparous animals, embryogenesis occurs in the absence of exogenous nutrients, making the maternal nutrients stored in the oocytes, such as yolk granules, critically important [8,155]. Oocyte maturation occurs during the preovipositional phase, characterized by a rapid increase in ovarian size [156,157,158,159]. It is during this phase that oocytes exhibit rapid growth due to the accumulation of RNA, carbohydrates, lipids, and proteins. These biomolecules will fulfill the regulatory and metabolic needs of the developing embryo, as well as support hatching and molting, until the embryo can feed on its first prey [54,91,160,161,162,163]. The absence of any yolk component can restrict or even block embryo development [14].

Most spiders protect their eggs in some type of silk sac (ootheca) [109], which serves to ensure adequate humidity, thermal insulation, and protection against parasites and microbes [164]. Egg development within this structure is divided into two stages. The first is the embryonic stage, which spans from fertilization until the eggs hatch inside the ootheca. Descriptions of spider embryonic development are often confusing, as different authors use varying terms for different embryonic stages (e.g., prelarvae, larvae, pronymphs, or embryos) [165,166,167,168,169]. The second stage is the post-embryonic stage, which includes the period from hatching to the emergence of juveniles from the ootheca. The duration of this emergence process varies between species, ranging from hours to several days [110,165,170,171,172].

Descriptions of spider embryonic development mainly focus on the morphological aspects of the process, such as the formation of the embryonic rudiment, morphology of cells in the extra-embryonic region, and the transition from radial to bilateral symmetry, among others [168,169,173,174,175,176,177,178,179,180,181,182,183,184,185]. However, information on biochemistry and energy metabolism during embryonic and post-embryonic development remains scarce.

Energy resource usage and the duration of embryonic development vary greatly among taxa, ranging from 30 h to over 3 months [186]. Although lipids are the primary energy source in insects, crustaceans, and mites [119,187,188,189,190,191], the embryo’s high energy demand could also be met by proteins and carbohydrates [188,191,192]. In spiders, yolk depletion is a critical point, as it can lead to competition for prey or cannibalism [193]. Furthermore, the consumption of lipoproteins during embryonic development is also an important energy resource [16,54,194].

To date, information on energy resource consumption during embryonic and post-embryonic development is available only for the species *P. pythagoricus* and *P. saltans*. In these species, the roles of TAGs, carbohydrates, and proteins have been analyzed [54,110], as well as the consumption of residual lipoprotein reserves after emergence [194]. In these studies, the authors identified five intra-oothecal stage, of which the first three correspond to embryonic stages, with a duration of 10 days for *P. pythagoricus* and 15 days for *P. saltans*. The remaining two stages are post-embryonic, each lasting 15 days for both species. Additionally, three extra-oothecal stages for *P. pythagoricus* were studied, lasting 17 days, which include the stages of gregarious juveniles and dispersed juveniles [54,110] (Figure 1).

During the early developmental stages, in *P. pythagoricus*, there was no variation in the total protein concentration, but a gradual consumption of lipoproteins was observed (Figure 2). A similar pattern has been described in other arthropods, including certain insects and mites, where protein content remains largely unchanged during embryogenesis [188,195,196,197,198]. However, in the spider *P. saltans*, a gradual consumption of proteins was observed, even though lipoprotein levels did not decrease during embryonic development [54].

In *P. pythagoricus*, high-molecular-weight proteins (120 and 75 kDa) from lipoprotein were consumed during post-embryonic stages, whereas lower molecular weight proteins (46 and 30 kDa) were utilized after juvenile emergence. A similar pattern was observed in the crustacean *Macrobrachium borellii*, where the lipoprotein contained a larger subunit (which surrounded smaller proteins) that was more susceptible to enzymatic attack [48]. Conversely, the 67 kDa yolk protein (homologous to Hc) persisted throughout all developmental stages in *P. pythagoricus*. Following oothecal emergence, *P. pythagoricus* juveniles exhibited increased total protein consumption and a 24.4% depletion of LP reserves, likely due to their intense activity, particularly during dispersal stages. It appears to be common for spiderlings to emerge with yolk reserves, as what was observed in *P. pythagoricus* aligns with what happens in *P. saltans*, where dispersal begins with 24% of the yolk remaining [194].

Irie and Yamashita [199] reported that in *Bombyx mori*, Vn degradation occurs only during the last days of embryonic development, and newly hatched larvae have around 30% of the initial Vn content of the eggs. However, Oliveira et al. [200] found that *Rhodnius prolixus* nymphs hatch with 50% of the initial Vn, while *Boophilus microplus* larvae retain approximately 60% [63].

As previously mentioned, the presence of Hc in the eggs of *P. pythagoricus* is likely due to the incorporation of maternal hemolymph Hc by the oocytes. Thus, this Hc could provide the embryo with an initial pool of this protein to meet its needs (oxygen transport and storage) until it can synthesize its own [109,201]. Although early embryonic Hc synthesis cannot be ruled out, it is likely that higher expression levels occur at later developmental stages, as described for some members of the Subphylum Chelicerata and other arthropods [146,202,203]. Leite et al. [204] studying embryos 80 h post egg-laying in the spider *P. tepidariorum*, observed the expression of Hc-related genes in a cell type likely corresponding to hemocytes. The increase in Hc concentration during postembryonic development (Figure 2) could allow juveniles to acquire new tools to cope with their environment after emergence, as Hc has a wide range of functions (in addition to those previously mentioned), including participation in pathogen defense, hormone transport during molting, lipid transport, and phenoloxidase activity associated with cuticle sclerotization. The latter function is particularly important during the development of *P. pythagoricus*, as the cuticle pigmentation process begins just before the emergence of the egg sac [110]. Finally, it is possible that, in the early stages of development, maternal Hc fulfills the embryo’s oxygen requirements, since the high rate of aerobic metabolism in embryonic development generates a significant demand for oxygen [203,205,206]. In more advanced stages, when cellular differentiation becomes more evident, embryonic Hc production may begin.

Carbohydrates are essential for energy metabolism. They associate with Vgs to form glycoproteins, which are stored in the yolk and actively contribute to embryogenesis [54,188,191,192]. In some arthropods, carbohydrate mobilization for energy production during embryo development has been observed [54,188,191,192,207]. Glycogen stored in the eggs is typically used consistently during embryonic development, but in some arthropods (e.g., *Drosophila melanogaster*), it has been observed that its content increases in the later embryonic stages during organogenesis [208,209]. A similar pattern likely occurs in *P. pythagoricus*, where glycogen concentration increases at later developmental stages (Figure 3). Different carbohydrate sources, such as glycogen, may be important for chitin biosynthesis in later developmental stages because large amounts of glucose are required [210]. In *P. pythagoricus*, it was observed that carbohydrates during the post-embryonic period were consumed in a manner similar to that in *P. saltans*, demonstrating a gradual and consistent consumption throughout post-embryonic and post-emergence stages. However, these egg reserves of *P. pythagoricus* do not represent more than 1%, and combined with this result, it has been described that the glycosylation of LV in this species was minimal, representing less than 2% of the m/m ratio [57] (Table 4). This percentage likely explains the small amount of carbohydrates present in the eggs during embryonic stages (only 0.3% of their mass). In contrast, *S. malitiosa* exhibits a higher LV glycosylation percentage (3.6%), with total egg carbohydrate content reaching 17% of their mass [55], while the spider *P. saltans* shows an intermediate level with values of 2.7% of its mass [54]. In this latter case, glucose content in developing eggs was studied, showing an increase in late embryonic periods, likely due to the increased activity of the embryo, which requires rapid mobilization of energy components.

The knowledge of energy resources supporting embryonic and post-embryonic development in spiders is limited, likely because studies on spiders, as in many other arthropods, have primarily focused on Vgs/LVs as the only or primary source of yolk energy [44,48,114,116,212]. However, evidence suggests that spider eggs, such as those of *S. malitiosa*, contain a significant proportion of lipids that are not associated with LVs [98]. On the other hand, while numerous studies have examined the role of different biomolecules during development in insects and crustaceans, only a few have done so in spiders. Comparing and interpreting these studies is challenging because each presents results in a different manner. Synthesizing all of these results to make generalizations about the process becomes a significant challenge [190,191,213,214].

In oviparous species, yolk lipids serve as one of the primary energy sources. In insects and arachnids, it has been observed that lipid content in larvae and juveniles is higher than in adults of the same species. It is estimated (in some species) that if these energy reserves were maintained until adulthood, they could compensate for the total energy required for adult life [215]. Similarly, during *P. pythagoricus* development, intra-oothecal stages and newly emerged gregarious juveniles exhibited higher lipid concentrations than both dispersed juveniles and adults.

Over time, the importance of energy lipids as a key resource for embryonic development has been emphasized in various insects, mites, and crustaceans [119,187,188,189,190]. However, this pattern appears to differ in some spider species. For instance, *P. pythagoricus* eggs initially contain 72.2% structural lipids and only 24.3% energy lipids, corresponding to approximately 13.3 μg of energy lipids per mg of wet weight. A similar composition has been reported for *S. malitiosa*, whose eggs contain 67% structural lipids (PE + PC + SM) and only 26.6% energy lipids (TAG + free fatty acids (FFAs)) [55]. In contrast, in another species of the Lycosidae family, *P. saltans*, energy lipids are predominant, accounting for 51.5% of total lipids, followed by 33.1% of structural lipids [54]. According to the authors, differences in the lipid composition of spider eggs may be related to variations in feeding strategies, as previously observed in the hemolymph lipoproteins of *Aphonopelma californicum* and *Brachypelma albopilosum* (Araneae: Theraphosidae) [216].

Although phospholipids were previously thought to have an exclusively structural role, the consumption of PE during the postembryonic period provides strong evidence that a significant portion of PE may also serve an energy function. This aligns with previous suggestions that certain structural lipids can be utilized as an energy source [129,130,217], as observed for phospholipids in crustaceans [131,218]. This hypothesis is further supported by the low PE content found in late-stage juveniles and adults (0.4% and 0.21% of the initial PE content, respectively), indicating that, after hatching, PE would primarily undergo hydrolysis during the postembryonic period. Moreover, phospholipids may also play regulatory roles. The hydrolysis of PE can provide ethanolamine, which is involved in protein modifications [219], while PC, which is consumed during postembryonic development, may supply choline, a key component in methyl group metabolism and neurotransmission [220]. The substantial consumption of phospholipids during postembryonic stages suggests that phospholipase activity is likely higher than lipase activity.

In contrast, the concentration of SM remained constant throughout development, suggesting a predominantly structural role. The combination of different lipids is essential for maintaining membrane integrity [221,222], and contributes, at least in part, to membrane rigidity [223,224].

In *P. pythagoricus*, the content of TAG decreased during the dispersal stages, with nearly 50% being consumed after the first molt outside the ootheca (stage 7) and almost entirely depleted one week later (stage 8) [110]. A similar pattern of late-stage TAG hydrolysis has been observed in *P. saltans* and some crustaceans [54,225]. This pattern of consumption contrasts with that described in embryos of other invertebrates, where 40–60% of TAG is utilized during early development [226,227,228]. Although the possibility of de novo synthesis of FFA cannot be ruled out, the observed increase in FFA concentration in gregarious juveniles (stage 6) is likely due to the hydrolysis of phospholipids, primarily PE. In contrast, in dispersed juveniles, this increase could result from TAG hydrolysis.

In spiders, hydrocarbons may serve various functions, including acting as the first chemical barrier against pathogen entry, serving as kairomones for entomopathogenic fungi and bacteria [229], contributing to water homeostasis regulation [230,231], and playing an active role in chemical communication between conspecifics [153,232]. In *P. pythagoricus*, hydrocarbon content varies significantly, particularly during specific embryonic stages and in dispersed juvenile stages. This may be due to de novo synthesis, as described in insects [233], where increases in hydrocarbon content were observed during early developmental stages and the first nymphal stage. In contrast, cholesterol (COL) levels remain relatively stable across the analyzed stages, despite the key role of COL and EE as membrane components and precursors of molting hormones. Furthermore, hydrocarbons, COL, and EE play a fundamental biological role in the formation of ecdysteroids [234,235,236].

Nutritional requirements during the early stages of embryonic development differ among species [237], reflecting different types of diets and metabolic adaptations [238,239] and in many cases, they are species-dependent [238]. However, in scorpions and some spiders, it was observed that any variations caused by different diets are not significant or do not notably affect the FA composition, unlike what has been described for other arthropods [101,102,103]. Eggs of *P. pythagoricus* mainly contain unsaturated fatty acids (UFAs), with 45.94% of 18:1 and 23.36% of 18:2, and saturated fatty acids (SFAs), with 14.55% of 16:0 and 9.52% of 18:0. These major FAs are stored in the ovaries during vitellogenesis and subsequently incorporated into the yolk, aligning with the general FA profile of LVs [57]. This FA pattern was also observed in two members of the Lycosidae family, *S. malitiosa* and *P. saltans*, species that exhibited 18:2, 18:1, 16:0, and 18:0 as the major FAs [98,180]. While some studies indicate that FA composition remains unchanged throughout embryonic development [240], in certain insects, and in other arthropods, FA consumption appears to be selective, with some FAs being metabolized while others accumulate during embryogenesis [241]. In spiders, the total percentage of SFA, monounsaturated fatty acids (MUFAs), and polyunsaturated fatty acids (PUFAs) remained constant until the juveniles emerged from the ootheca (stage 6). However, after emergence, the percentage of MUFAs decreased while PUFA increased in the dispersed juveniles, accompanied by a specific change in FA composition within these groups. The depletion of 16:0 is balanced by an increase in 18:0, explaining the overall stability of SFA levels throughout development, particularly in the dispersed stages, where TAG consumption is evident.

In addition to being an important energy source, derivatives of 16:0, such as 14-methylhexadecanoic acid, have been identified in silk, along with other lipids exhibiting significant antimicrobial activity due to the protective effect of the methyl group [242,243]. This is particularly relevant given that active silk production begins as soon as juveniles emerge from the ootheca. Regarding UFAs, declines in 18:1 and an increase in 18:2 were observed, with these FAs playing an important role in cellular physiology, immunity, and reproduction [244]. In dispersed *P. pythagoricus* juveniles, the content of 20:4 and 20:5 fatty acids increased six-fold compared to the intraoothecal stages. These FAs are considered essential for arthropods because they cannot synthesize them [245,246,247]. The significance of 20-carbon FAs lies in their role as precursors of eicosanoids, including prostaglandins and leukotrienes [248,249,250,251,252]. Numerous studies on arthropods highlight the critical function of eicosanoids in immune defense against microorganisms [249,253,254], which is vital for the survival of newly dispersed spider juveniles.

## 4. General Conclusions

This review compiles the available information to date on a topic of great importance for arachnids as well as any oviparous animal: reproduction. In recent years, significant efforts have been made to understand the morphological and physiological changes in spider eggs using specialized high-resolution techniques such as X-ray microtomography, SEM equipped with software for 4D recording, fluorescent dyes, single-cell sequencing, and SPLiTseq scRNA-seq [22,169,204,255]. However, information regarding the biochemistry and energy metabolism of reproduction in general, and specifically the biochemistry of vitellogenesis and embryogenesis, remains limited.

The lipoprotein structures of Vg and LV are compared across different arachnid groups, as well as their possible function and relationship. Information concerning the two proteins linked to Vg (30 and 47 kDa) common to some spider species is presented and discussed. The IGS and IHS indexes are introduced for the first time in spiders, providing a new tool for studying vitellogenesis.

The lipid dynamics between the various organs involved in vitellogenesis (MD, hemolimph and ovary) are described. Studying females in different reproductive stages such as previtellogenesis, early vitellogenesis, vitellogenesis, and postvitellogenesis, as well as males, it was possible to observe how lipids and FAs are synthesized, mobilized, and accumulated in the yolk. Structural lipids such as PE and PC are primarily stored in the yolk, followed by TAGs and other lipids. During embryonic development, lipids are consumed differentially, with structural lipids being consumed in the early stages and energetic lipids in the later stages.

The presence of Hc is described for the first time in early spider eggs, with its function seemingly linked to the initial requirements of organogenesis. An increase in Hc concentration was observed in the advanced stages of embryonic development. Finally, the consumption of LV was observed gradually throughout embryonic development, with a small percentage remaining as an energy reserve in the solitary juvenile.

The data reviewed in this manuscript represent the first advances in understanding the reproductive process, both in arthropods in general and in arachnids in particular. The uptake and transfer of nutrients from mothers to their offspring is a key aspect of development and reproductive success in organisms. The studies analyzed in this review are of great importance for future research in various areas related to spiders, such as their adaptability and evolution.

## Figures and Tables

**Figure 1 insects-16-00398-f001:**
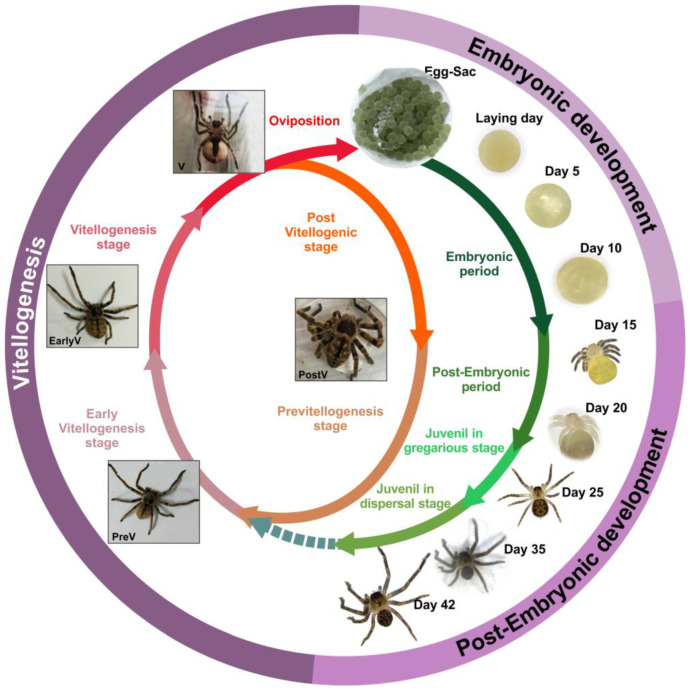
Representative diagram of the study of the reproductive cycle of *P. pythagoricus*. Vitellogenic, embryonic and post-embryonic development.

**Figure 2 insects-16-00398-f002:**
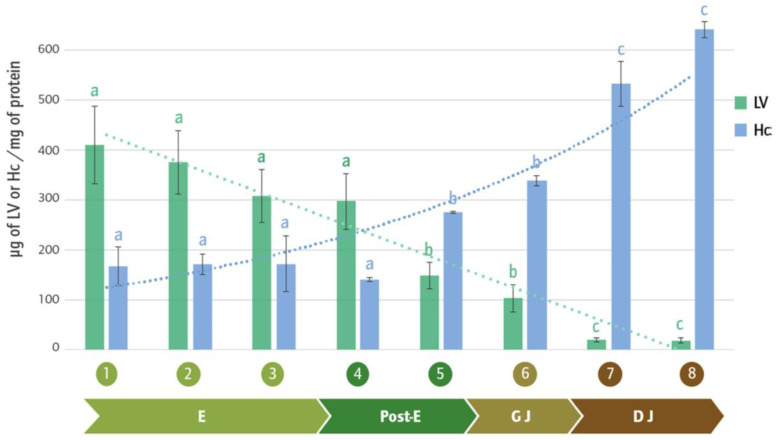
Quantitative analysis of LV and Hc present in the development of *P. pythagoricus*. Values are the means ± SDs. Different letters above the bars indicate significant differences among the different stages at *p* < 0.05, as determined by Tukey’s post hoc test. 1 to 8: stages of *P. pythagoricus* development; E: embryonic period; Post-E: post-embryonic period; GJ: juveniles in gregarious stage; DJ: juveniles in dispersal stages.

**Figure 3 insects-16-00398-f003:**
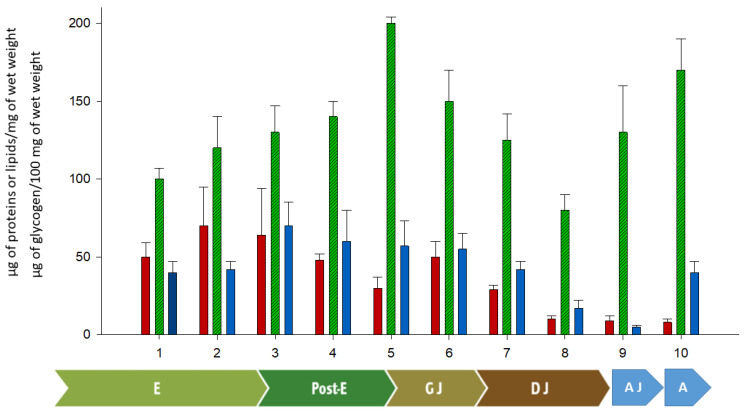
Quantitative analysis of lipids (red), proteins (green) and glycogen (blue) present in the development of *P. pythagoricus.* Values are the means ± SDs. 1 to 8: stages of *P. pythagoricus* development; E: embryonic period; Post-E: post-embryonic period; GJ: juveniles in gregarious stage; DJ: juveniles in dispersal stages, AJ: advanced juvenile stage, A: adult stage.

**Table 1 insects-16-00398-t001:** Comparative table of Vns and Vgs of spiders and acari members.

Species of Arachnids	Number of Majority Eggs Proteins and Their Molecular Weight (kDa)	Presence of Vgs	Vg Origin or Detection Place
*Tegenaria atrica* [18] (spider)	47-	x	Detection: hemolymph and ovaries.
*Schizocosa malitiosa* [55] (spider)	SmLV1: 116, 87 and 42 SmLV2: 135, 126, 109 and 70	-	Detection: eggs
*Pardosa saltans* [54] (spider)	116, 87, 70 and 42	-	Detection: eggs
*Polybetes pythagoricus* [56,57] (spider)	LV1: 120, 75, 46 and 30 LV2: 170, 120, 109, 75, 67 and 30	-	Detection: eggs
*Parasteatoda tepidariorum* [16] (spider)	Vg 250, 47 and 30		Origen: midgut glands, ovaries and hemolymph.
*Ixodes scapularis* [58,59] (acari)	154, 135, 87, 78, 67, 64 and 35	x	Origin: fat body Detection: fat bod and hemolymph
*Ornithodoros moubata* [41] (acari)	160, 140, 125, 100, 64 and 50	x	Detection: fat body, midgut and hemolymph
*Argas hermanni* [60] (acari)	Exogenous: 66.2 to >200 Endogenous: 22 to 59 Eggs-specific 35.5 and 47.2	x	Detection: hemolymph and ovary
*Ornithodoros parkeri* [61] (acari)	160, 140, 125, 100, 64 and 50	x	Origin: fat body Detection: fat body and hemolymph
*Dermacentor variabilis* [62] (acari)	VnA: 135, 110, 98, 80, 67, 50, 45 and 35 VnB: Identical to VnA with the addition of 93 kDa subunit.	x	Origin: fat body and midgut Detection: fat body and midgut, hemolymph, and ovary
*Rhipicephalus microplus* (formerly *Boophilus microplus*) [63,64,65,66] (acari)	107, 102, 87, 67, 65 and 44	Multiples Vgs	Origin: ovaries and extraovarian tissues. Detection: hemolymph and ovary
*Tetranychus urticae* [67,68] (acari)	(Multiple subunits) 3.6 to 290	-	-
*Haemaphysalis longicornis* [29,69] (acari)	203, 147, 126, 82, 74, 70, 61, 47 and 31	Multiples Vgs	Detection: fat body, hemolymph, and ovary

**Table 2 insects-16-00398-t002:** N-terminal amino acid sequence of subunits of LV or Vg and genes encoding Vg.

Species of Arachnids	Molecular Weight	N-Terminal Sequence	Detection Place	Observations
*P. pythagoricus* [56]	75 kDa	AEKMADW(S)KYLKE	Egg	
	67 kDa	VVKEKEDRILEXFE	Egg	
	46 kDa	SIMYNEKDDIXVENR	Egg	
	30 KDa	(G)PFQRQSQXAT(R)	Egg	
*Tegenaria atrica* [18]	47 kDa	XVEDIEGEVQERLRE	Hemolymph and ovarian	
*Pardosa pseudoannulata* [70]				cDNAs encoding vitellogenins (PpVg1, PpVg2 and PpVg3)
*Parasteatoda tepidariorum* [16]				Two genes encoding Vg (PtVg4 and PtVg6)

**Table 4 insects-16-00398-t004:** Amount of proteins, lipids and carbohydrates of Vg and yolk of spiders, insects and crustaceans.

	Protein (µg/mg Eggs)			Lipid (µg/mg Eggs)			Carbohydrates (µg/mg Eggs)			References
	LV1	LV2	Total	LV1	LV2	Total	LV1	LV2	Total	
*Polybetes pythagoricus* (spider)	13.1	11.1	87.5	13.2	1.2	50	0.3	0.08	0.3 (glycogen)	[57,110]
*Schizocosa malitiosa* (spider)	3.9	3.3	12.6	1.6	0.3	8.1	0.1	0.2	4.5	[55]
*Pardosa saltans* (spider)	7.7		22.1						0.8	[54]
*Adalia bipunctata* (insect)			130			100			1.8	[211]
*Adalia decempunctata* (insect)			145			115			2.4	[211]
*Anisosticta novemdecimpunctata* (insect)			145			125			1.3	[211]
*Macrobrachium borellii* (crustacean)			9.2			2.7				[48,119]

## Data Availability

The original contributions presented in this study are included in the article. Further inquiries can be directed to the corresponding author.

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
