# Peer review of "Vitellogenesis and Embryogenesis in Spiders: A Biochemical Perspective"

_insects, 2025, doi:10.3390/insects16040398_

Round 1
Reviewer 1 Report
Comments and Suggestions for Authors
Comments on the manuscript "Vitellogenesis and embryogenesis in spiders: a biochemical perspective". The topic is interesting and, as the authors mention on several times, current knowledge about of these chemical aspects of the early developmental stages of arachnids is very limited.
The review covers two main topics: vitellogenesis and embryonic development in spiders, highlighting similarities and differences with other arthropod invertebrates. This adds considerable value to the review presented by the authors.
The manuscript is generally well structured and written. I have few minor suggestions, which are detailed below.
Figure 1: I think the caption should describe the content of the figure more accurately.
Table 3: Is "egg" shown as a percentage? It would be good to clarify this (%). Also, decimal separators should be standardised, using either points or commas consistently.
Finally, after reading the manuscript, a question arises: How might this information be relevant to other areas of spider biology? Do the authors think that there could be variations between different groups of spiders that could help to elucidate their evolutionary history? A short paragraph addressing these questions might increase interest in studying these processes.
Author Response
Comment 1: The manuscript is generally well structured and written. I have few minor suggestions, which are detailed below.
Figure 1: I think the caption should describe the content of the figure more accurately.
Table 3: Is "egg" shown as a percentage? It would be good to clarify this (%). Also, decimal separators should be standardised, using either points or commas consistently
Answer1: The legend for Figure 1 was fixed and the tables are now corrected.
Comment 2: Finally, after reading the manuscript, a question arises: How might this information be relevant to other areas of spider biology? Do the authors think that there could be variations between different groups of spiders that could help to elucidate their evolutionary history? A short paragraph addressing these questions might increase interest in studying these processes.
Answer 2: A new paragraph regarding this was added.
Reviewer 2 Report
Comments and Suggestions for Authors
This manuscript fills a gap in the study of the reproductive biochemistry of spiders. Despite the wide variety of spider species, biochemical research on their reproduction remains relatively limited. This paper provides valuable information in the field of spider reproduction by synthesizing the biochemical processes from vitellogenesis to postembryonic development in spiders. Apart from that, it compares the structure, function, and relationship of vitellogenins and lipovitellins in different species, revealing the synthesis, mobilization, and accumulation of lipids and fatty acids during the vitellogenesis process. The article also explores some molecular interactions involved in spider embryonic development.
One advantage of this article is that it provides new tools for studying spider reproduction. It introduces biometric parameters, such as gonadosomatic and hepatosomatic index in spider study for the first time. It is an excellent paper and provides important scientific insights of spider reproduction and developmental mechanisms, fills a research gap in the related field, and inspires new ideas and methods for future studies.
The main problem is that the hormonal regulation during the gonadotrophic cycle is missing, which is a little disappointing. Second, the authors use peptide to describe Vg (30 and 47 kDa) is not very suitable. Usually, for the molecule larger than 5 KDa, we call it protein. And insulin is the smallest protein. So peptides should be replaced by protein, when it refers to the large molecules such as Vg or lipovitellin.
Author Response
Comments 1: One advantage of this article is that it provides new tools for studying spider reproduction. It introduces biometric parameters, such as gonadosomatic and hepatosomatic index in spider study for the first time. It is an excellent paper and provides important scientific insights of spider reproduction and developmental mechanisms, fills a research gap in the related field, and inspires new ideas and methods for future studies.
The main problem is that the hormonal regulation during the gonadotrophic cycle is missing, which is a little disappointing
Response: Although there is very little information on the subject, a paragraph was added.
Comments 2: Second, the authors use peptide to describe Vg (30 and 47 kDa) is not very suitable. Usually, for the molecule larger than 5 KDa, we call it protein. And insulin is the smallest protein. So peptides should be replaced by protein, when it refers to the large molecules such as Vg or lipovitellin.
Response 2: The word polypeptide was replaced by protein in several places.